# DIFFERENTIABLE LEARNING OF NUMERICAL RULES IN KNOWLEDGE GRAPHS

**Po-Wei Wang**
Machine Learning Department
Carnegie Mellon University and
Bosch Center for AI
poweiw@cs.cmu.edu

**Daria Stepanova**
Bosch Center for AI
daria.stepanova@de.bosch.com

**Csaba Domokos**
Bosch Center for AI
csaba.domokos@de.bosch.com

**Zico Kolter**
Department of Computer Science
Carnegie Mellon University and
Bosch Center for AI
zkolter@cs.cmu.edu

## ABSTRACT

Rules over a knowledge graph (KG) capture interpretable patterns in data and can be used for KG cleaning and completion. Inspired by the `TensorLog` differentiable logic framework, which compiles rule inference into a sequence of differentiable operations, recently a method called Neural LP has been proposed for learning the parameters as well as the structure of rules. However, it is limited with respect to the treatment of numerical features like *age*, *weight* or *scientific measurements*. We address this limitation by extending Neural LP to learn rules with numerical values, *e.g.*, "*People younger than 18 typically live with their parents*". We demonstrate how dynamic programming and cumulative sum operations can be exploited to ensure efficiency of such extension. Our novel approach allows us to extract more expressive rules with aggregates, which are of higher quality and yield more accurate predictions compared to rules learned by the state-of-the-art methods, as shown by our experiments on synthetic and real-world datasets.

## 1 INTRODUCTION

Due to the availability of vast amounts of knowledge on the web, advances in information extraction have led to large graph-structured knowledge bases, also known as *knowledge graphs* (KGs), which are widely used in web search, question answering, and data analytics. Such KGs represent data as a graph of entities (*e.g.*, $john$, $article1$) connected via relations (*e.g.*, $citedIn$), or more formally as a set of binary grounded atoms (*e.g.*, $citedIn(john, article1)$). A common task in such settings is that of *link prediction*, determining whether a relation exists between two entities in the graph even if the relation is not included explicitly in the graph. Although most work on this topic has focused on statistical rule-extraction techniques (Meilicke et al. (2019); Galárraga et al. (2015); Ortona et al. (2018a)), recent methods have shown the benefit of using deep learning approaches for this link prediction task (see Wang et al. (2017) for overview). And while most deep approaches (for example, those based upon graph embedding methods) are inherently difficult to interpret, the Neural LP method of Yang et al. (2017) is particularly appealing in that it allows for interpretable

resulting rules for the link prediction task while still preserving the flexibility of a learning approach. Unfortunately, Neural LP is also quite limited in the types of rules it is capable of representing, and notably no rules that depend on *numerical features* can be efficiently learned within this framework.

In this paper, we propose an extension to Neural LP that allows for fast learning of numerical rules. Specifically, although numerical rules would result in dense matrix operations in the generic Neural LP framework, we show that using dynamic programming and cumulative sum operations, we can efficiently express the operators for numerical comparators within the Neural LP framework. By defining the relevant operators implicitly in this manner, we show that we can extend Neural LP to efficiently learn rules that make use of numerical features, while retaining the interpretability of the Neural LP framework. More generally, this is an instance of integrating so-called "aggregates" (*i.e.* external oracle queries, in this case binary queries that reflect numerical comparison) within a rule-learning framework. Learning such rules with aggregates is very much an open problem in the KG community (Galárraga & Suchanek (2014)), and our approach is the first work to learn rules with these numerical aggregates.

We apply our approach to several knowledge graph datasets, and show that we are able to answer queries more accurately than the previous Neural LP approach, as well as more accurately than a state-of-the-art rule extraction method, the AnyBurl package proposed by Meilicke et al. (2019). Specifically, we show on two synthetic and two real-world datasets that our extension to Neural LP is able to more accurately recover rules that depend on numerical information, and thus make much more accurate link predictions in the knowledge graph. Further, the extracted rules are still interpretable as in the original Neural LP framework, and unlike the pure graph embedding strategies (Bordes et al. (2013)).

## 2 RELATED WORK

**Relational Data Mining**. The problem of learning rules from the data has been traditionally addressed in the area of relational data mining Raedt (2017) and inductive logic programming (ILP) Muggleton (1995). Works most related to ours concern learning decision trees with aggregates Vens et al. (2006) from relational data. However, these methods typically do not scale well, and modern knowledge graphs are far beyond what they can handle.

In the context of KGs, the problem of rule learning has recently gained a lot of attention. In Ortona et al. (2018b) rules with negation, which also support numerical comparison as we do have been considered. Contrary to our approach, however, Ortona et al. (2018b) is designed to find a small set of rules that cover the majority of positive and as few negative examples as possible, which differs from our goal of learning rules in an unsupervised fashion.

**Neural-based Rule Learning**. Several works Yang et al. (2017); Manhaeve et al. (2018); Rocktäschel & Riedel (2017); Evans & Grefenstette (2018); Zhang et al. (2019); Ho et al. (2018); Weber et al. (2019) utilize embedding models and neural architectures for rule learning. The closest to ours is the work Yang et al. (2017), which reduces the rule learning problem to algebraic operations on neural-embedding-based representations of a given KG. However, Yang et al. (2017) is restricted to non-numerical rules in contrast to our work.

**Embedding Models with Numerics**. The problem of KG incompleteness has been tackled by methods that predict missing relational edges between existing entities. Several approaches rely on statistics and include tensor factorization (*e.g.*, Nickel et al. (2011)). Other models are based on neural embeddings (*e.g.*, Bordes et al. (2013)). For overview see Wang et al. (2017).

The most relevant for us is the work García-Durán & Niepert (2018), which presents a novel approach to combining relational, latent (learned) and numerical features, *i.e.* features taking large or infinite number of real values for the KG completion task. While García-Durán & Niepert (2018) operates on KGs with numerical values, it's results like in the case of most knowledge graph embedding models are not interpretable.

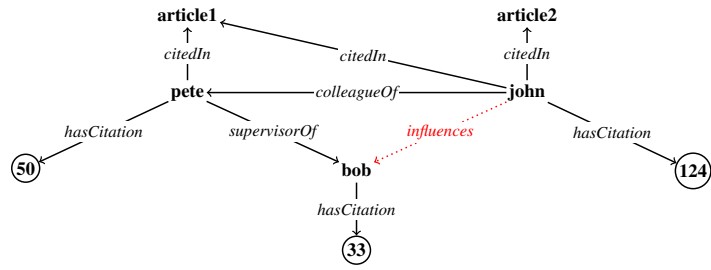

Figure 1: An exemplar KG about publications, their authors and relations among them. Relations are presented in italics, entities in bold black and numerical values in circes, true facts as solid black lines and the missing ones as dashed lines in red.

## 3 PRELIMINARIES

**Knowledge Graphs**. We assume countable sets $\mathcal{C}$ of constants (*a.k.a.* entities), $\mathcal{N} \subset \mathbb{R}$ of numerical values and $\mathcal{R}$ of binary relations (*a.k.a.* predicates). A KG $\mathcal{G}$ is defined by a finite set of ground atoms (*a.k.a.* facts), of the form $p(x,y)$, where $p \in \mathcal{R}$, $x \in \mathcal{C}$ and $y \in \mathcal{C} \cup \mathcal{N}$ (*e.g.*, $citedIn(john, article1)$). The set $\mathcal{R}_n \subseteq \mathcal{R}$ stores all numerical predicates $p$, such that $p(x,y) \in \mathcal{G}$, where $x \in \mathcal{C}$ and $y \in \mathcal{N}$. The set of numerical facts, *i.e.* facts over numerical predicates, is denoted by $\mathcal{G}_n \subseteq \mathcal{G}$. We use lower-case letters for constants and upper-case letters for variables.

As KGs are incomplete, one can assume that missing facts, *i.e.* facts that are not in $\mathcal{G}$, are either *unknown* or *false*. Typically the *open world assumption* (OWA) is employed, which means that missing facts are considered to be unknown rather than false. Alternatively, the *local closed world assumption* (LCWA) can be considered to generate negative facts by assuming that the KG is locally complete as data is usually added to KGs in batches. More precisely, it means that for any $x \in \mathcal{C}$ we can conclude that $p(x,y)$ is false if $z \in \mathcal{C} \cup \mathcal{N}$ exists such that $p(x,z) \in \mathcal{G}$ and $p(x,y) \notin \mathcal{G}$.

**Numerical Rules**. A *rule* is an expression of the form

$$p(X,Y) \leftarrow q_1(X,Z_1) \wedge \ldots \wedge q_n(Z_n, Y) , \tag{1}$$

where $p, q_1, \ldots, q_n \in \mathcal{R}$, left-hand side of the rule is referred to as the rule head, and right-hand side as the rule body, and every conjunct in the rule head or body is referred to as an atom. The rule $influences(X,Y) \leftarrow colleagueOf(X,Z) \wedge supervisorOf(Z,Y)$ intuitively states that typically students are influenced by colleagues of their supervisors. Aside from conventional rules, we can also have *numerical rules*, *i.e.* rules that contain numerical *comparison* among variables (*e.g.*, number of citations of two people), or a variable and a numerical constant.

To simplify presentation, numerical values in $\mathcal{N}$ linked to an entity from $\mathcal{C}$ are sometimes treated as its "features", and numerical relations in $\mathcal{R}_n$ as functions that depend on those features. In this case, for $p(X,Y)$ we also use a shortcut notation $X.p = Y$. For instance, $john.hasCitation = 124$ stands for $hasCitation(john, 124)$, and for compactness, $r^\circ_{pq}(X,Y)$ stands for $X.p \circ Y.q$, where $\circ \in \{\leq, >\}$. The second subscript in $r^\circ_{pq}$ is omitted if it is clear from the context that $p = q$.

**Example 1.** *For example, consider a KG in Figure 1 and the following rule*

$$influences(X,Y) \leftarrow colleagueOf(X,Z) \wedge supervisorOf(Z,Y) \wedge r^>_{hasCitation}(X,Z) .$$

*This rule states that students are influenced by colleagues of their supervisors with a higher number of citations*[1].

We can also define a classification relation mapping the feature to the probability of a logistic classification, $\sigma(w^T X.features + b)$, where $w$ and $b$ are parameters and $\sigma$ is the sigmoid function. As we demonstrate later such rules can be integrated and learnt naturally in the Neural LP framework.

**Rule Learning**. Given a KG $\mathcal{G}$ the goal of *rule learning* is to extract rules from $\mathcal{G}$, such that their application to $\mathcal{G}$ results in an approximation of the ideal KG, which stores all correct facts.

---

[1]*I.e.* $influences(X,Y) \leftarrow colleagueOf(X,Z) \wedge supervisorOf(Z,Y) \wedge hasCitation(X,V_1) \wedge hasCitation(Z,V_2) \wedge V_1 > V_2$.

The Neural LP method (Yang et al. (2017))is among rule learning proposals, which learns a distribution over rules of the form in Eq. (1) without comparison operators in an end-to-end fashion by making use of gradient-based optimization. This approach relies on the `TensorLog` framework (Cohen et al. (2017)), which connects rule application with sparse matrix multiplications. In `TensorLog` all entities are mapped to integers, and each entity $i$ is associated with a one-hot encoded vector $v_i \in \{0,1\}^{|\mathcal{C}|}$ such that only its i-th entry is 1.

For example, every KG entity $c \in \mathcal{C}$ in Fig. 1 is encoded as a 0/1 vector of length 5, since $|\mathcal{C}| = 5$. For every relation $p \in \mathcal{R} \setminus \mathcal{R}_n$ and every pair of entities $x, y \in \mathcal{C}$ a matrix $M_p \in \{0,1\}^{|\mathcal{C}| \times |\mathcal{C}|}$ is defined such that its $(y, x)$ entry, denoted by $(M_p)_{yx}$, is 1 iff $p(x, y) \in \mathcal{G}$. For example, by considering the KG in Fig. 1, for the relation $p = citedIn$ we have

$$
M_p = \begin{array}{c} \\ \\ \\ \\ \\ \end{array} \begin{array}{ccccc} \text{john} & \text{pete} & \text{bob} & \text{article1} & \text{article2} \\ \left[ \begin{array}{ccccc} 0 & 0 & 0 & 0 & 0 \\ 0 & 0 & 0 & 0 & 0 \\ 0 & 0 & 0 & 0 & 0 \\ 1 & 1 & 0 & 0 & 0 \\ 1 & 0 & 0 & 0 & 0 \end{array} \right] & & & & \end{array} \begin{array}{c} \text{john} \\ \text{pete} \\ \text{bob} \\ \text{article1} \\ \text{article2} \end{array}
$$

The idea of `TensorLog` is to imitate application of rules for any entity $X = x$ by performing matrix multiplications $M_{q_n} \dots M_{q_2} M_{q_1} v_x = s$, where $v_x$ is the indicator of entity $x$. Non-zero entries in the vector $s$ point to the entities $y$ for which $p(x, y)$ is derived by applying the above rule on $\mathcal{G}$. For example, the inference for the following rule $influences(X, Y) \leftarrow colleagueOf(X, Z), supervisorOf(Z, Y)$ can be translated to

$$
M_{supervisorOf} \, M_{colleagueOf} \, v_x = s \, .
$$

By setting $v_x = [1, 0, 0, 0, 0]^\top$ as indicator of $john$ and applying the matrix multiplications, we obtain $s = [0, 0, 1, 0, 0]^\top$, the indicator of $bob$. As $M_{q_1}, \dots, M_{q_n}$ are sparse, the matrix-vector multiplication can be done efficiently, and the inference process is parallelizable on GPUs.

In Neural LP (Yang et al. (2017)) above operators are used to learn for every head the formula

$$
f(\alpha) = \sum_i \alpha_i \prod_{j \in \beta_i} M_{q_j} \tag{2}
$$

where $i$ indexes over all possible rules, $\alpha_i$ is the confidence associated with the rule $r_i$ and $\beta_i$ is an ordered list of all relations appearing in these rules. The rules are read off from the solution of the following optimization problem

$$
\max_{\{\alpha_i, \beta_i\}} \log \left( \sum_{\{x, y\}} v_y^\top \left( \sum_i \alpha_i \left( \prod_{j \in \beta_j} M_{q_j} \right) v_x \right) \right) \, .
$$

## 4 LEARNING RULES WITH NUMERICAL FEATURES AND NEGATIONS

As the main contribution of the paper we extend the Neural LP framework to allow us to use comparison operators with numerical values in the rule bodies, and also to handle negations of atoms. These extensions are non-trivial as the Neural LP framework does not directly support facts over numerical values: naively treating numerical constants as entities in $\mathcal{C}$ is intractable due to the explosion of the number of non-zero elements in the respective matrices. Similarly, naive treatment of negated atoms would introduce dense matrices that would not be practical to operate on. Intuitively, the main idea of our approach is to represent the necessary matrix operations implicitly, either using dynamic programming, cumulative sums and permutations (for numerical comparison features) or low rank factorizations (for negated atoms). Exploiting this structure lets us formulate the associated `TensorLog` operators efficiently, and effectively integrate them into the Neural LP framework for rule extraction.

### 4.1 COMPARISON OPERATORS

**Pair-wise Comparison**. We start by implicitly representing the operators associated with numerical comparators. Let $p, q \in (\mathbb{R} \cup \{\texttt{NaN}\})^{|\mathcal{C}|}$ be the vector of two specific features, where $\texttt{NaN}$ means missing values. The comparison operator $M_{r_{\tilde{p}q}^{\leq}}$ is defined as

$$(M_{r_{\tilde{p}q}^{\leq}})_{ij} = \begin{cases} 1 & \text{if } p_i \leq q_j \text{ and } p_i, q_j \text{ are not } \texttt{NaN}, \\ 0 & \text{otherwise.} \end{cases}$$

Intuitively, this matrix includes the binary indicator of the comparison, over all pairs of entities in the knowledge graph that contain $p$ and $q$. Unlike conventional sparse relations, the matrix $M_{r_{\tilde{p}q}^{\leq}}$ is usually dense (*i.e.* it has $O(n^2)$ non-zero elements), thus a naive materialization would exceed the typical GPU memory limit. However, in reality there is no need to explicitly materialize the `TensorLog` relation matrix. Note that in the Neural LP inference chain we described above, all that is needed is to efficiently compute the matrix-vector product between a relation matrix and some vector representing the current probabilities in the inference chain.

Consider the special case that both $p$ and $q$ are sorted in an ascending order as $\tilde{p}$ and $\tilde{q}$ with operator $(\leq)^2$, and call the corresponding comparison matrix $\tilde{M}_{r_{\tilde{p}q}^{\leq}}$. Since $\tilde{q}_i \leq \tilde{q}_{i+1}$ and $\tilde{p}_j \leq \tilde{p}_{j+1}$, we have the following property (**P1**)

$$(\tilde{M}_{r_{\tilde{p}q}^{\leq}})_{i,j} = 0 \implies (\tilde{M}_{r_{\tilde{p}q}^{\leq}})_{i+1,j} = 0$$
$$(\tilde{M}_{r_{\tilde{p}q}^{\leq}})_{i,j} = 1 \implies (\tilde{M}_{r_{\tilde{p}q}^{\leq}})_{i,j+1} = 1,$$

i.e., the resulting matrix $\tilde{M}_{r_{\tilde{p}q}^{\leq}}$ is always effectively *lower triangular* in form (or more precisely, the transition from 1 to 0 is always monotonic in the matrix, even if the non-zero pattern is not precisely lower triangular in the usual sense).

Now define $\gamma_i = \arg\max_j$ such that $(\tilde{M}_{r_{\tilde{p}q}^{\leq}})_{ij} = 1$, i.e., $\gamma_i$ is the index of the last element equal to one. The main observation is that we can compute the required matrix-vector product using just this $\gamma$ vector, *i.e.* for any vector $v$,

$$(\tilde{M}_{r_{\tilde{p}q}^{\leq}} v)_i = \sum_{\gamma_i \leq j \leq |\mathcal{C}|} v_j = \texttt{cumsum}(v)_{\gamma_i} \, .$$

The respective values of $\gamma$ for $\tilde{M}_{r_{\tilde{p}q}^{\leq}} v$ can be precomputed on a CPU with linear complexity by dynamic programming since its value is monotonically increasing because of the property (**P1**). Also, the `cumsum` operator can be calculated in $O(|\mathcal{C}|)$ time, with an efficient GPU parallelization that in practice is even faster for large vectors.

For the general case when $p$ and $q$ are not sorted, we can first permute the input $v$ to the sorted order, perform the matrix-vector multiplication, then permute the result back to the original order. Since permutation (*a.k.a.*, index slicing) is a simple linear time operation, this does not affect the complexity of the overall approach. Specifically, let $P_p$ and $P_q$ be the permutation matrices corresponding to the argsort of $p$ and $q$, respectively. Then the matrix-vector multiplication corresponding to the comparison operator can be written as

$$M_{r_{\tilde{p}q}^{\leq}} v = (P_q^{\top} \underbrace{P_q) M (P_p^{\top}}_{\tilde{M}} P_p) v = P_q^{\top} (\tilde{M} P_p v) = P_q^{\top} \texttt{cumsum}(P_p v)_{\gamma} \, ,$$

which can be computed in $O(|\mathcal{C}|)$ in parallel given $\beta$, which are precomputed once in $O(|\mathcal{C}| \log |\mathcal{C}|)$. Thus, the comparison operator needed for inference can be computed efficiently on a GPU.

**Efficient Use of Numerical Comparisons via Multi-atom Symbol Matching**. Although the above numerical comparison operator provides an efficient means for implementing such comparisons within the Neural LP framework, it has significant drawbacks as well. Specifically, because the comparison operator is dense, when using it to match potential entities in the graph, it has the

---

[2] This way, the comparison involving $\texttt{NaN}$ always yields false and they will be stacked at the end.

potential to create a huge number of candidate matches. For example, the operator $(X.p \leq Y.p)$ will link the entity with smallest $p$ to all other entities with attribute $p$ and will decrease the probability of finding the correct target. To make the comparison operator more useful, it is natural to use it jointly with some other sparse operator. For example,

$$colleagueOf(X,Y) \wedge (X.p \leq Y.q)$$

would search only over neighbors of $X$ in the graph that also obey the respective numerical relation. Let the two parallel relations from above correspond to operators $M_{colleagueOf}$ and $M_{r_{\bar{p}q}^{\leq}}$ respectively. The above conjunction can be implemented in `TensorLog` via

$$(M_{colleagueOf}v) \odot (M_{r_{\bar{p}q}^{\leq}}v) = \text{diag}(M_{colleagueOf}v)(M_{r_{\bar{p}q}^{\leq}}v),$$

where the symbol $\odot$ denotes the element-wise multiplication. Unfortunately, the above relation is not learnable in the standard Neural LP framework, since the latter only supports a single chain of matrix-vector operations $M_{q_n} \dots M_{q_2} M_{q_1} v_x$, which does not allow for easy computation of this Hadamard product, as it includes two "copies" of the vector $v$. However, we note that it is trivial to simply cache intermediate values of $v$ in the multiplication chain, and this way conveniently compute such Hadamard products; in the knowledge graph setting, this exactly corresponds to the ability to integrate symbol matching at multiple points in the inference chain.

**Classification Operators**. We may also consider more general rules, where the comparison is performed not necessarily among two numerical attributes of a certain entity but rather functions over such attributes. Note that such comparison for all entities can readily be expressed by `TensorLog` operators, that is, the corresponding matrix for a given numerical value $Z$ is a diagonal matrix. We model the numerical value $Z$ by making use of a logistic model. Namely, for each entity we collect the feature vector $\varphi$, which consists of all the numerical values from $\mathcal{N}$ that are in relation with the given entity. The $i^{\text{th}}$ element of the diagonal in $M$ is defined as $\text{sigmoid}(w^\top \varphi + b)$, where the weight vector $w$ and the bias vector $b$ are assumed to be learned. These parameters can easily be learnt in the Neural LP framework via backpropagation.

**Negated Operators**. The negation of a relation $p \in \bar{\mathcal{R}}$ obtained by naively flipping all zeros to ones and vice versa in the corresponding (sparse) matrix $M_p$ results in a dense matrix, which is not supported directly in `TensorLog`. To compute the negated operator $\bar{M}_p \in \{0,1\}^{|\mathcal{C}| \times |\mathcal{C}|}$ we employ the *local closed-world assumption*. For a given $M_p$ only the elements, that are in such rows that contain at least one non-zero element, should be flipped. The matrix-vector multiplication for the negated operator $\bar{M}_p$ can be written as

$$\bar{M}_p v := 1_p \mathbf{1}^\top v - M_p v , \tag{3}$$

where $1_p \in \{0,1\}^{|\mathcal{C}|}$ is the indicator vector for $p$ such that $(1_p)_i = 0$ iff $p_i = $ `NaN`. Here, $\mathbf{1}$ is the vector with all of its elements equal to 1. Note that for any `TensorLog` operator $M_p$ the products $M_p v$ and $1_p(\mathbf{1}^\top v)$ can be computed efficiently, therefore the negated operator $\bar{M}_p$ can be computed efficiently as well.

The trick in Eq. (3) generalizes to the comparison operators $M_{r_{pq}^\circ}$, namely,

$$\bar{M}_{r_{pq}^\circ} v = 1_p 1_q^\top v - M_{r_{pq}^\circ} v .$$

E.g., $\bar{M}_{r_{\bar{p}q}^{\leq}} v = 1_p 1_q^\top v - M_{r_{pq}^{>}} v$. This way we can learn rules with negated atoms in the body.

Once the rules have been learned by our approach, we rely on the same procedure as in Yang et al. (2017) to decode them back to the form of Eq. 1.

**Connection to Rules with Aggregates over Knowledge Graphs.** Note that, importantly, the rules that we extract using the described procedure fall into the language of logic rules with external computations in the spirit of Eiter et al. (2012), and are connected to the concept known as *aggregates* in the knowledge representation community. Indeed, much of the formulation we have presented here can be viewed as an instance of learning rules with aggregates from knowledge graphs. This is an active area of current research, and our work here is significant in connection to this area in that we present one of the first methods for learning rules using (a limited form of) such aggregates. However, the discussion requires substantial additional notation in order to be concrete, and so we defer this discussion to Appendix A

| Dataset | $|\mathcal{C}|$ | $|\mathcal{R}|$ | $|\mathcal{R}_n|$ | $|\mathcal{G}_n|$ | $|\mathcal{G}|$ | $|\mathcal{G}_t|$ |
|---|---|---|---|---|---|---|
| FB15K-237Num | 12493 | 237 | 116 | 27899 | 82992 | 10359 |
| DBP15K | 12867 | 278 | 251 | 48105 | 79345 | 9789 |
| Numerical1 | 1000 | 2 | 1 | 1000 | 5785 | 98 |
| Numerical2 | 1000 | 3 | 2 | 2000 | 5800 | 100 |

Table 1: Dataset statistics, where $\mathcal{G}_t$ stands for the KG corresponding to the testset.

## 5 EXPERIMENTAL RESULTS

In this section we report the results of our experimental evaluation, which focuses on the effectiveness of our method against the state-of-art rule learning systems with respect to the predictive quality of the learned rules. Specifically, we conduct experiments on a canonical knowledge graph completion task as described in Yang et al. (2017). In this task, the query and tail are given to the algorithm, and the goal is to retrieve the related head. For example, if $supervisorOf(turing, church)$ is not present in the knowledge graph, then when presented with the relation $supervisorOf$ and the entity $church$, the goal is to exploit the existing triples in the KG to retrieve $turing$. In order to represent the query as a continuous input to the neural controller, for each query we learn the embedding of the lookup table. As in Yang et al. (2017), the embedding has dimension 118 and is randomly initialized to unit norm vectors. The only difference between the parameters of the Neural-LP and our system is that we set the learning rate to $10^{-2}$, while in Yang et al. (2017) it is set to $10^{-3}$, but both systems are run to convergence, and this learning rate does not affect the final performance materially except for making it converge faster. In all cases, we extracted rules with a maximum length of 5.

### 5.1 EXPERIMENTAL SETUP

**Datasets**. To evaluate and compare our approach for learning numerical rules, we considered the following datasets containing knowledge graphs:

- *FB15K-237-num* is a variant of Freebase knowledge graph with numerical values, where the reverse relations have been removed (García-Durán & Niepert (2018)).
- *DBPedia15K* is a fragment of the DBPedia knowledge graph Lehmann et al. (2015) restricted to numerical facts García-Durán & Niepert (2018).
- *Numerical1* is a synthetic dataset with 1000 entities, each containing a single numerical value (generated uniformly from 1 to 1000). Each entity has 50 randomly-chosen neighbors, and the goal is to find neighbors with the closest value to the current entity given the constraint that the neighbor's value must be higher.
- *Numerical2* is a variant of the *Numerical1* dataset, where each entity has two numerical values, "balance" and "debt"; under the same generation process as above, the goal is to find a neighbor of each node with the largest delta between balance and debt.

The statistics of the knowledge graphs used in our experiments is presented in Table 1, where apart from the number of KG entities ($|\mathcal{C}|$), facts ($|\mathcal{G}|$) and the size of the test set ($\mathcal{G}_t$), we also report the number of numerical relations ($|\mathcal{R}_n|$) and numerical facts ($|\mathcal{G}_n|$). We use 80% of the KG as the training set, and 10% for test set and the same for validation. The KG is split randomly with the constraint that only non-numerical facts appear in the test set, since we do not learn rules capable of predicting missing facts over numerical entities.

**Baselines**. We compared our proposed approach, which we refer to as *Neural-Num-LP* against the following two baselines:

- *AnyBURL*[3] (Meilicke et al. (2019)) is an anytime bottom-up method for learning Horn rules, *i.e.* rules with only positive atoms and no comparison operators. To tune the system we use the default parameters as described on the system webpage and set the timeout to 5000 seconds.
- *Neural-LP*[4] (Yang et al. (2017)) is a differential rule learning system described in Section 3.

---

[3]http://web.informatik.uni-mannheim.de/AnyBURL/
[4]https://github.com/fanyangxyz/Neural-LP

| Dataset | Numerical1 | | Numerical2 | | FB15K-237-num | | DBP15K-num | |
|---|---|---|---|---|---|---|---|---|
| | Hit@10 | MRR | Hit@10 | MRR | Hit@10 | MRR | Hit@10 | MRR |
| AnyBurl | 0.031 | 0.009 | 0.685 | 0.509 | **0.426** | 0.244 | 0.522 | 0.371 |
| Neural-LP | 0.240 | - | 0.295 | - | 0.362* | 0.240* | 0.436 | - |
| Neural-Num-LP | **1.000** | **0.941** | **1.000** | **0.837** | 0.415 | **0.259** | **0.682** | **0.451** |

Table 2: Comparing our approach against current state-of-the-art rule learning methods. * annotated entries obtained from Yang et al. (2017).

$$
\begin{aligned}
r^1: \quad & prefer(X, Y) \leftarrow neighbour(X, Y) \wedge r^{>}_{hasOrder}(X, Y) \wedge f(Y) \\
r^2: \quad & prefer(X, Y) \leftarrow neighbour(X, Y) \wedge f(Y) \wedge hasBalance(Y, Z_1) \wedge borrowed(Y, Z_2) \\
r^3: \quad & symptomHasRiskFactors(X, Y) \leftarrow f(X) \wedge symptomOfDisease(X, Z) \wedge \\
& \qquad\qquad\qquad\qquad\qquad diseaseHasRiskFactors(Z, Y) \\
r^4: \quad & defends(X, Y) \leftarrow ministerOfDefense(X, Z) \wedge f(Z) \wedge militaryBranchOfCountry(Z, Y)
\end{aligned}
$$

Table 3: Example rules generated by Neural-LP-N on the DBPedia15K knowledge graph. See text for a discussion of these rules.

Following the common practice Meilicke et al. (2019); Yang et al. (2017) we compute the standard evaluation metrics used for the link prediction task Bordes et al. (2013): *Hit@10*, the number of correct head terms predicted out of the top 10 predictions; and mean reciprocal rank (*MRR*), the mean of one over the rank of the correct answer in the list of predictions. We have implemented our approach for learning numerical rules from knowledge graphs in python using the PyTorch library, and conducted all experiments on a machine GTX 1080 TO GPU with 11 GB RAM.

## 5.2 RESULTS

In Table 2 we report the quality of predictions obtained by our method and the baselines. Since the Neural-LP framework Yang et al. (2017) cannot handle the Freebase with numerical information, we present the results for *FB15K-237* without numerical facts instead, which are taken from Yang et al. (2017). The MRR values are missing for Neural-LP in several places, as the implementation provided by the authors does not have the respective function implemented.

As expected, on the synthetic datasets *Numerical1* and *Numerical2*, our method significantly out-performs the baselines. This is natural, since these datasets are constructed so that reasoning about numerical attributes is required for almost any prediction task presented to the algorithms. And notably, the proposed approach is able to achieve 100% Hit@10 rates, as it is able to correctly identify these relevant numerical properties. This contrasts to the baseline Neural-LP approach, which is unable to incorporate such information, and thus predicts the heads of each relation more or less randomly. The datasets are also particularly challenging for AnyBURL, because AnyBURL treats each numerical value as an independent entity, and thus cannot perform efficient comparative reasoning.

Most compellingly, however, similar observations can be made about the real-world datasets as well. Indeed, since all entities in the KG including numerical ones are treated equally by the available systems, intuitively both AnyBurl and Neural-LP try to find frequent patterns in KGs, and use these to predict the missing facts. The numerical rules mined by our system are much more expressive and substantially improve the performance of the approach in some cases. Specifically, our method outperforms the Neural-LP approach in terms of all metrics on both the *FB15K-237-num* and *DBPedia15K* datasets. The AnyBURL dataset is still competitive with our approach on the *FB15K-237-num* dataset (better in terms of Hit@10 but worse in terms of MRR), but our approach substantially outperforms it on the *DBPedia15K*, where our reasoning involving numerical comparison is able to substantially improve upon the existing methods.

**Examples of extracted rules**. In Figure 3 we present examples of the rules learned by our system. In particular, the rules $r^1$, and $r^2$ have been extracted from the datesets *Numerical1* and *Numerical2* respectively, the rule $r^3$ from *FB15K-237-num* and the rest of the rules from *DBPedia15K*. For example, $r^1$ is the rule with a comparison operator, which states that a person $X$ prefers neighbours with the maximal order that is less than $X$'s. The rule $r^3$ reflects that symptoms with certain properties (described by the function $f$) typically provoke risk factors inherited from diseases which have these symptoms. Here, the function $f$ is the sigmoid over a linear combination of numerical

properties of $X$. Finally, $r^4$ states that prime ministers of countries with certain numerical properties (described by the function $f$), are supported by military branches of the given country. Here, the function $f$ is again the sigmoid over a linear combination of numerical properties of $Y$.

## 6 Conclusion

In this paper we have addressed the problem of learning numerical rules from large knowledge graphs. Especially we have considered rules, where in the rule bodies numerical comparison operators and aggregates (*i.e.* external oracle queries) that enable us to aggregate numerical properties of entities are allowed. The Neural-LP method is a recent appealing learning approach based on `TensorLog`; however it does not support numerical rules, as they would result in dense matrix operations. We have introduced an extension to the Neural-LP framework that allows for learning such rules from KGs by efficiently expressing comparison and classification operators, negation as well as multi-atom symbol matching. We have shown that our proposed extension outperforms previous techniques that do not support numerical information with respect to the quality of predictions that they produce. The future research might focus on a further extension of our current approach by allowing for more general rule forms with complex external computations as well as rules with existential variables in the head.

### Acknowledgments

Authors would like to thank Christian Meilicke for his support with adjusting the parameters of the AnyBurl package as well as anonymous reviewers for their invaluable feedback.

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

## A  INTERPRETATION AS RULES WITH EXTERNAL COMPUTATIONS

We have presented our approach for learning rules that support numerical comparison among entities and classification of entities based on aggregation of their numerical features, and have also highlighted the algorithmic and numerical approaches to handling such rules.

In this Appendix we draw a connection between the rules of our focus and formalisms known in the knowledge representation community.

In the general case the rule features that we consider are inlined with *logic rules* that allow for *external computations*, *i.e.* "oracle programs" that might appear in the body of the rules as part of a generic black-box computation (in the spirit of answer set programs with external functions Eiter et al. (2012)). In the case of simplest numerical comparisons, for example, the corresponding oracle may check whether some numerical feature of one entity is larger than some numerical feature of another one. In a more complex setting an oracle may aggregate numerical features of an entity by computing their linear combination, and compare the result with a certain value.

Since the rules that we are targeting account both for relational and numerical information, they can be characterized as restricted explainable structures simultaneously supporting symbolic and sub-symbolic representations and inference. Learning such structures is a long-standing and important

goal of artificial intelligence, which has recently gained special attention (see e.g., Manhaeve et al. (2018); Weber et al. (2019); Rocktäschel & Riedel (2017); Evans & Grefenstette (2018)).

The challenge of learning such rules with aggregates is also a central one in much work on KGs Galárraga & Suchanek (2014). Indeed, discovering patterns by learning rules from knowledge graphs enables us to obtain concise descriptions of a domain as well as to complete and clean the data. Especially by considering numerical values such as age, weight or experimental measurements, which are common, e.g., for scientific knowledge graphs Auer et al. (2018), learned rules might reveal interesting correlations in the data or even lead to scientific discoveries.

Our proposed procedure is the first approach that is capable of learning such rules (even though we of course do not allow for arbitrary aggregates, the fact that we incorporate such learning at all is a substantial contribution to the KG literature). Because this is a key capability of our approach, in this section we further formally present the exact structure of the rules that we support and illustrate them by examples, motivating their applicability in various scenarios. The notation and definitions here are not crucial for understanding the remainder of the paper, but will provide context and clarify the contribution of the current work from the perspective of the knowledge representation community.

## A.1 RULES WITH AGGREGATES

We consider *aggregate atoms* of the form $f\{Y_1, \ldots, Y_n : p_1(X, Y), \ldots, p_n(X, Y)\} \circ Z$, where $f$ is an aggregate function symbol, $\{Y_1, \ldots, Y_n : p_1(X, Y), \ldots, p_n(X, Y)\}$ is called an aggregate element, $\circ \in \{\leq, >\}$ and $Z$ is a numerical value. For any aggregate in the above form we assume that $p_1, \ldots, p_n \in \mathcal{R}_n$, i.e., $p_1, \ldots, p_n$ are numerical features of an entity $X$.

**Example 2.** *The aggregate element* $\{Y_1, Y_2 : citedIn(X, Y_1), singleAuthorArticles(X, Y_2)\}$ *denotes the numerical feature vector containing* $\varphi = (Y_1, Y_2)$, *where* $Y_1$ *is the number of citations* $X$ *has, and* $Y_2$ *is the number of articles with* $X$ *being a single author.*

The function $f$ can in principle be arbitrarily complex and can be even represented by a neural network. In our work, as a starting point we restrict ourselves to linear functions of the following form: sigmoid($w^\top \varphi + b$), where $\varphi$ is the respective numerical feature vector for the target entity $X$.

**Example 3.** *Reconsider the knowledge graph in Figure 1 extended with the facts* $singleAuthorArticles(pete, 6), singleAuthorArticles(john, 20)$. *For the aggregate* $f\{Y_1, Y_2 : hasCitation(X, Y_1), singleAuthorArticles(X, Y_2)\} > 0.5$, *with* $f$ *being the sigmoid function of the above form, suppose that* $w = (1, 10)$ *and* $b = 200$. *We have* $\varphi = (50, 6)$ *for* $X = pete$ *and* $\varphi = (124, 8)$ *for* $X = john$. *Since* $50 + 6 \cdot 10 - 200 < 0.5$, *for pete the respective aggregate atom evaluates to false, while for* $john$ *it is true, which is easy to verify based on analogous computations.*

In this work we have described procedures for learning *rules* that have the following form

$$p(X, Y) \leftarrow q_1(X, Z_1) \wedge \ldots \wedge q_n(Z_n, Y). \tag{4}$$

where $p_1(Y, X)$ is an *atom*, and every $q_n(Y, Z_n), \ldots, q_1(Z_1, X)$ is either an *atom*, a *negated atom*, an *aggregate atom* of the form $f\{Y_1, \ldots, Y_m : p_1(X_1, Y_m), \ldots, p_m(X_1, Y_m)\} \circ 0.5$, $\circ \in \{\leq, >\}$ with $p_1, \ldots p_m \in \mathcal{R}_n$ and $f$ being a sigmoid function or it is a *comparison conjunction* of the form

$$r_{pq}^\circ(Z, Y) \leftarrow p(Z, N) \wedge q(Y, M) \wedge N \circ M.$$

where $\circ \in \{\leq, >\}$ and $Y, Z$ are present in some rule body atom elsewhere.

As described in Section 4 the above condition that $Y, Z$ must be connected via some relation ensures that the entities that participate in the comparison are semantically related to each other, which is needed to avoid irrelevant comparisons.

**Example 4.** *Consider the following rule*

$$topStudentOf(Z, Z') \leftarrow f\{Y_1, Y_2 : hasCitation(X, Y_1), singleAuthorArticles(X, Y_2)\} > 0.5 \wedge$$
$$supervisorOf(X, Z) \wedge affiliatedWith(X, Z').$$

*Intuitively, the aggregate function computes a certain index for a person based on the number of his citations and the number of publications, with him being a single author. The rule states that people*

*with an index above a certain threshold supervise only top students of the university with which they are affilliated.*

*The following expression demostrates rule with a comparison atom:*

$$influences(X, Y) \leftarrow colleagueOf(X, Z) \wedge supervisorOf(Z, Y) \wedge$$
$$hasCitation(X, N) \wedge hasCitation(Z, M), N > M.$$

*This rule states that students are typically influenced by colleagues of their supervisors with a higher number of citations. The expression $r_{hasCitation}^{\leq}(Y, Z)$ can be used as a shortcut for the last three atoms in the body of the above rule.*

## A.2 EXECUTION OF RULES WITH AGGREGATES

The execution of rules with negation and aggregates that we focus on is defined in the standard way (see Faber et al. (2011); Eiter et al. (2012) for details). More precisely, let $\mathcal{G}$ be a KG, $r$ a rule over $\mathcal{G}$, and $a$ be a standard atom from $\mathcal{G}$. Then, $rule \models_{\mathcal{G}} a$ holds if there is a variable assignment that maps atoms in the positive part of rule body ($body_{rule}^{+}$) in $\mathcal{G}$ such that it does not map any of the atoms in the negative part of rule body ($body_{rule}^{-}$) in $\mathcal{G}$. For aggregate atoms $a$ we have that $rule \models_{\mathcal{G}} a$ if the value of the respective aggregate function on the KG satisfies the given inequality constraint.

$\mathcal{G}_r = \mathcal{G} \cup \{a \mid r \models_{\mathcal{G}} a\}$ extends $\mathcal{G}$ with atoms derived from $\mathcal{G}$ by applying $r$. Note that to avoid propagating uncertain predictions, given a set of rules $R$ we execute every rule in $R$ on $\mathcal{G}$ independently, *i.e.* $\mathcal{G}_R = \bigcup_{r \in R} \mathcal{G}_r$. Given additional syntactic restrictions on rules in $R$, which disallow cycles through negation consistency is ensured.

