# OpenReview forum: "Differentiable learning of numerical rules in knowledge graphs"
_ICLR.cc/2020/Conference — Accept (Poster)_

### Official Review · AnonReviewer3 · 2019-10-22
**Official Blind Review #3**

**Rating:** 3

**Review:**

This paper proposes an interesting extension to the Neural LP framework for learning numerical rules in knowledge graphs. The proposed method can handle predicates involving the comparison of the numerical attribute values. The authors demonstrate its effectiveness on both synthetic knowledge graphs and the parts of existing knowledge graphs which consider numerical values.

I recommend the paper to be rejected in its current form for the following 3 reasons:

(1) Although the idea of making numerical rules differentiable is interesting, the current proposed method can only deal with one form of numerical predicate, which is numerical comparison. The limitation to such a special case makes the paper somewhat incremental.

(2) The paper does not do a great job of convincing the reader that the problem it is trying to solve is an important matter, or the proposed method is indeed effective in some applications. Although the proposed method does a good job in synthetic experiments, outperforming existing methods by a large margin, its performance on the numerical variants of Freebase/DBPedia dataset does not show consistent significant improvement. The authors should try to find a real-world domain which can really demonstrate the effectiveness of the method.

(3) The experiment section lacks more detailed analysis which can intuitively explain how well the proposed method performs on the benchmarks. A good place to start with is to visualize(print out) the learned numerical rules and see if they make any sense. The experiment section needs significant improvement, especially when there is space left.


The authors can consider improving the paper based on the above drawbacks. I encourage the authors to re-submit the paper once it's improved.


**Experience Assessment:**

I have read many papers in this area.

**Review Assessment: Checking Correctness Of Derivations And Theory:**

I carefully checked the derivations and theory.

**Review Assessment: Checking Correctness Of Experiments:**

I carefully checked the experiments.

**Review Assessment: Thoroughness In Paper Reading:**

I read the paper thoroughly.

---

> ### Author Response · Authors · 2019-11-14
> **Response to AnonReviewer3**
>
> We appreciate the comments of the reviewer. Please see our reply below.
>
> 1) - "... the current proposed method can only deal with one form of numerical predicate, which is numerical comparison."
>
> Apart from simple numerical comparison we are also able to deal with complex classification operators that aggregate numerical attributes using linear functions, where the threshold value is selected in a systematic fashion, (see Classification Operators) as well as negated atoms (see Negated Operators on p. 6). We note that such rules are indeed limited to some extent, but they still capture a rather expressive fragment of answer set programs with restricted forms of external computations [Eiter et al., 2012].
> Below we present examplar rules learned by our framework, which are not restricted to numerical comparisons.
>
> 2a) - "The paper does not do a great job of convincing the reader that the problem it is trying to solve is an important matter, or the proposed method is indeed effective in some applications."
>
> With the rapid development of industrial and scientific knowledge graphs, we believe (and agree with the Reviewer #2) that learning rules that involve multiple modalities is an important and relevant problem. Indeed, such rules can not only be used for data cleaning and completion, but they are also themselves extremely valuable assets carrying human-understandable structures that support both symbolic and subsymbolic representations and inference.
>
> 2b) -  "The authors should try to find a real-world domain which can really demonstrate the effectiveness of the method."
>
> To the best of our knowledge Freebase and DBPedia are the only standard KGs with numerical values [Garcia-Duran et al., 2018] used for the evaluation in state-of-the-art works. This is the reason why we have selected and used them for our experiments. The impact of our approach might appear to be rather modest, since these KGs still have only a limited amount of numerical information. Therefore, to demonstrate the power of our approach further, we have also performed evaluation on the synthetic datasets. We would be happy to learn about other datasets suitable for our experiments.
>
> 3) - "The experiment section lacks more detailed analysis which can intuitively explain how well the proposed method performs on the benchmarks. A good place to start with is to visualize (print out) the learned numerical rules and see if they make any sense."
>
> According to the Reviewer's comment we will extend Section 5 on experimental results by showing more detailed analysis. In particular, we will present the following examples of the learned rules from the considered (real-world and synthetic) datasets:
>
> - FB15K:
> 	disease_has_risk_factors(X,Z) :- f(X), symptom_of_disease(X,Y), disease_has_risk_factors(Y,Z)
> The rule states that symptoms with certain properties (described by the function f) typically provoke risk factors inherited from diseases which have these symptoms. Here, the function f is the sigmoid over a linear combination of numerical properties of X.
>
> - DBPedia:
> 	defends(X,Z) :- primeMinister(Z,Y), militaryBranch(Y,X), f(Y)
> This rule states that prime ministers of countries with certain numerical properties (described by the function f), are supported by military branches of the given country. The function f is the sigmoid over a linear combination of numerical properties of Y.
>
> - Numerical1:
> 	prefer(X,Y) :- isNeighbourTo(X,Y), hasOrder(X,Z1), hasOrder(Y,Z2), Z1>Z2, max{Z2:hasOrder(Y,Z2)}
> This rule with a comparison operator states that a person X prefers neighbours with the maximal order that is less than X's.
>
> - Numerical2:
> 	prefer(X,Y) :- isNeignborTo(X,Y), hasBalance(Y,Z1), borrowed(Y,Z2), f(Y)
> This rule states that neighbours with the largest difference between the balance and the borrowed amount are preferred.
> More precisely, here f selects among all X those entities, for which the difference between the balance and the borrowed amount is maximal.

---

> > ### Comment · AnonReviewer3 · 2019-11-15
> > **Thanks for the response**
> >
> > Thanks for the response! However, the response does not resolve my concern about whether the task is significant enough in practice. Also the experiment part is not updated accordingly. Therefore I will not change the rating.

---

### Official Review · AnonReviewer1 · 2019-10-23
**Official Blind Review #1**

**Rating:** 6

**Review:**

This paper proposed several extensions to the Neural LP work. Specifically, this paper addresses several limitations, including numerical variables, negations, etc. To efficiently compute these in the original Neural LP framework, this paper proposed several computation tricks to accelerate, as well as to save memory. Experiments on benchmark datasets show significant improvements over previous methods, especially in the case where numerical variables are required.

I think overall the paper is written clearly, with good summarization of existing works. Also I like the simple but effective tricks for saving the computation and memory.

One main concern is, how general this approach would be? As it is a good extension for Neural LP, it is not clear that the framework of Neural LP is flexible or powerful enough in general. For example, if rules contain quantifiers, how would this be extended?

Minor comments:

1) 4.1,  “O(n^2/2)” -- just put O(n^2) or simply write as n^2/2.
2) How are the rules from in Eq (2)? i.e., how is \beta_i selected for each i? In the extreme case it would be all the permutations.
3) I would suggest a different name other than Neural-LP-N, as it is somewhat underselling this work. Also it makes Table 2 not that easy to read.


**Experience Assessment:**

I have read many papers in this area.

**Review Assessment: Checking Correctness Of Derivations And Theory:**

N/A

**Review Assessment: Checking Correctness Of Experiments:**

I assessed the sensibility of the experiments.

**Review Assessment: Thoroughness In Paper Reading:**

I read the paper thoroughly.

---

> ### Author Response · Authors · 2019-11-14
> **Response to AnonReviewer1**
>
> We appreciate the Reviewer's comments, which help us to improve the paper. In the final version of the paper we will take them into consideration. In the following we reply to the main concerns of the reviewer.
>
> Q1 - "... how general this approach would be? ...if rules contain quantifiers, how would this be extended?"
> The extendibility of the Neural LP framework is a very important and relevant question, which we also mentioned explicitly as a possible future work direction.
> In the rules that we support in our framework all variables are universally quantified. While learning rules with existential quantifiers in rule heads is a difficult endeavor in general, even for classical relational learners, the Neural LP framework in principle can be extended to support them as follows: For every relation p, we can create a fresh diagonal Boolean matrix $M_{\exists p}$, which has 1 at the position (i,i) iff there exists an entity j, such that p(i,j) is in the KG (similar as for classification operators discussed on p. 5). Incorporating these matrices into the framework and filtering rules that have the respective relations in the head should allow us to extract the target rules. Yet analysing how well such approach performs in practice is still an open problem, which we leave for future work. In any case, we will discuss the extendability of the framework in the paper.
>
> Minor comment 1) - 4.1,  "O(n^2/2) -- just put O(n^2) or simply write as n^2/2".
> This is correct, thank you. We will fix this in the final version.
>
> Minor comment 2) - "How are the rules from in Eq (2)? i.e., how is \beta_i selected for each i? In the extreme case it would be all the permutations."
>
> To avoid exponential enumeration of the predicate orderings sophisticated transformation of the rules has been applied in the Neural LP framework (see [Yang et al. 2017]).
>
> Minor comment 3) - "I would suggest a different name other than Neural-LP-N..."
> Thanks for this suggestion. We will certainly consider renaming the approach and fixing this in Table 2.

---

> > ### Comment · AnonReviewer1 · 2019-11-15
> > **RE: Response to AnonReviewer1**
> >
> > Thanks for the reply and also the comments about handling existential quantifiers. Given this, I think I'm fine with the limitations in the current work.

---

### Official Review · AnonReviewer2 · 2019-10-25
**Official Blind Review #2**

**Rating:** 6

**Review:**

This paper proposes an extension of NeuralLP that is able to learn a very restricted (in terms of expressiveness) set of logic rules involving numeric properties. The basic idea behind NeuralLP is quite simple: traversing relationships in a knowledge graph can be done by multiplicating adjacency matrices, and which rules hold and which ones don't can be discovered by learning an attention distribution over rules from data.

The idea is quite clever: relationships between numeric data properties of entities, such as age and heigh, can also be linked by relationships such as \leq and \geq, and those relations can be treated in the same way as standard knowledge graph relationship by the NeuralLP framework.

A major drawback in applying this idea is that the corresponding relational matrix is expensive to both materialise, and use within the NeuralLP framework (where matrices are mostly sparse). To this end, authors make this process tractable by using dynamic programming and by defining such a matrix as a dynamic computation graph by means of the cumsum operator.  Furthermore, authors also introduce negated operators, also by defining the corresponding adjacency matrices by means of computation graphs.

Authors evaluate on several datasets - two real world and two synthetic - often showing more accurate results than the considered baselines.


One thing that puts me off is that, in Table 2, AnyBurl (the single one baseline authors considered other than the original NeuralLP) yields better Hits@10 values than Neural-LP-N, but the corresponding bold in the results is conveniently omitted.

Another concern I have is that the expressiveness of the learned rules can be somehow limited, but this paper seems like a good star towards learning interpretable rules involving multiple modalities.


Missing references - authors may want to consider citing https://arxiv.org/abs/1906.06187 as well in Sec. 2 - it seems very related to this work.


**Experience Assessment:**

I have published one or two papers in this area.

**Review Assessment: Checking Correctness Of Derivations And Theory:**

I carefully checked the derivations and theory.

**Review Assessment: Checking Correctness Of Experiments:**

I carefully checked the experiments.

**Review Assessment: Thoroughness In Paper Reading:**

I read the paper thoroughly.

---

> ### Author Response · Authors · 2019-11-14
> **Response to AnonReviewer2**
>
> We are really thankful for the positive feedback. Here we give detailed answers to the Reviewer's concerns.
>
> 1) - "... in Table 2, AnyBurl ... yields better Hits@10 values than Neural-LP-N, but the corresponding bold in the results is conveniently omitted."
>
> Thanks for pointing this out! We will make the presentation of the results consistent by highlighting the respective number.
>
> 2) - "... the expressiveness of the learned rules can be somehow limited,..."
>
> We remark that our framework supports rules with negation, comparison among numerical attributes and classification operators, where linear functions over attributes can be expressed. Such rules capture a fragment of answer set programs, where a limited form of aggregation [Faber et al., 2011] and restricted external computation functions [Eiter et al., 2012] are allowed. While these rules might not cover all possible knowledge constructs, they are still valuable and rather expressive for encoding correlations among numerical and relational features. Moreover, to the best of our knowledge they have not been directly supported by previous works on rule learning.
>
> 3) - "Missing references - authors may want to consider citing https://arxiv.org/abs/1906.06187 ..."
>
> Thanks for referring us to this important work! We will certainly add this reference to the paper.

---

> > ### Comment · AnonReviewer2 · 2019-11-15
> > **Thank you**
> >
> > Thank you for answering -- about 2, I mean the extracted rules still do not support less trivial operations such as aggregations (sum, mean, ..) and math operations (such as the sum). There is some work investigating how to learn these using neural architectures, such as https://arxiv.org/abs/1808.00508 . It would have been great to see them in here but I understand it can be tricky.

---

### Decision · Program_Chairs · 2019-12-19

**Decision:**

Accept (Poster)

**Comment:**

This paper presents a number of improvements on existing approaches to neural logic programming. The reviews are generally positive: two weak accepts, one weak reject. Reviewer 2 seems wholly in favour of acceptance at the end of discussion, and did not clarify why they were sticking to their score of weak accept. The main reason Reviewer
 1 sticks to 6 rather than 8 is that the work extends existing work rather than offering a "fundamental contribution", but otherwise is very positive. I personally feel that
a) most work extends existing work
b) there is room in our conferences for such well executed extensions (standing on the shoulders of giants etc).

Reviewer 3 is somewhat unconvinced by the nature of the evaluation. While I understand their reservations, they state that they would not be offended by the paper being accepted in spite of their reservations.

Overall, I find that the review group leans more in favour of acceptance, and an happy to recommend acceptance for the paper as it makes progress in an interesting area at the intersection of differentiable programming and logic-based programming.